# Unwrap RAP1’s Mystery at Kinetoplastid Telomeres

**DOI:** 10.3390/biom14010067

**Published:** 2024-01-04

**Authors:** Bibo Li

**Affiliations:** 1Center for Gene Regulation in Health and Disease, Department of Biological, Geological, and Environmental Sciences, College of Arts and Sciences, Cleveland State University, 2121 Euclid Avenue, Cleveland, OH 44115, USA; b.li37@csuohio.edu; 2Case Comprehensive Cancer Center, Case Western Reserve University, 10900 Euclid Avenue, Cleveland, OH 44106, USA; 3Department of Inflammation and Immunity, Lerner Research Institute, Cleveland Clinic, 9500 Euclid Avenue, Cleveland, OH 44195, USA; 4Center for RNA Science and Therapeutics, Case Western Reserve University, 10900 Euclid Avenue, Cleveland, OH 44106, USA

**Keywords:** RAP1, telomere, antigenic variation, telomeric silencing, VSG monoallelic expression, *Trypanosome brucei*, telomere stability

## Abstract

Although located at the chromosome end, telomeres are an essential chromosome component that helps maintain genome integrity and chromosome stability from protozoa to mammals. The role of telomere proteins in chromosome end protection is conserved, where they suppress various DNA damage response machineries and block nucleolytic degradation of the natural chromosome ends, although the detailed underlying mechanisms are not identical. In addition, the specialized telomere structure exerts a repressive epigenetic effect on expression of genes located at subtelomeres in a number of eukaryotic organisms. This so-called telomeric silencing also affects virulence of a number of microbial pathogens that undergo antigenic variation/phenotypic switching. Telomere proteins, particularly the RAP1 homologs, have been shown to be a key player for telomeric silencing. RAP1 homologs also suppress the expression of Telomere Repeat-containing RNA (TERRA), which is linked to their roles in telomere stability maintenance. The functions of RAP1s in suppressing telomere recombination are largely conserved from kinetoplastids to mammals. However, the underlying mechanisms of RAP1-mediated telomeric silencing have many species-specific features. In this review, I will focus on *Trypanosoma brucei* RAP1’s functions in suppressing telomeric/subtelomeric DNA recombination and in the regulation of monoallelic expression of subtelomere-located major surface antigen genes. Common and unique mechanisms will be compared among RAP1 homologs, and their implications will be discussed.

## 1. The Telomere Structure and Telomere Functions

Telomeres are nucleoprotein complexes located at chromosome ends [1]. In most eukaryotes, the telomere contains simple repetitive sequences, such as (TTAGGG)_n_ in vertebrates, with the G-rich strand going 5′ to 3′ toward the chromosome end [2,3]. Although most parts of the telomere are double-stranded, the very end of the telomere has a single-stranded 3′ overhang [4,5]. The telomere 3′ overhang can fold back and invade the duplex telomeric DNA and form the T-loop structure, which has been observed in human, *Oxytricha fallax* (a hypotrichous ciliate), and *Trypanosoma brucei* (a kinetoplastid parasite) [6,7,8], while a similar loop structure of the telomere chromatin has also been observed in mouse and chicken [9].

Telomeres are essential for maintaining the linear genome stability [10]. They form a specialized structure that protects DNA ends from nucleolytic degradation, prevents natural chromosome ends from being recognized as DNA damage sites, and suppresses illegitimate processes, such as DNA recombination and end joining [11,12,13,14,15]. The T-loop structure can effectively sequestrate and bury the telomere end to protect it from degradation and DNA recombination [13,16]. In addition, telomere proteins play critical roles in chromosome end protection in all eukaryotes that have been investigated so far [17,18], even though telomere proteins are only partially conserved from protozoa to mammals [19,20]. Vertebrate telomeres have both Shelterin and CST protein complexes [21]. Shelterin contains six core telomere proteins: TRF1 and TRF2 bind the duplex TTAGGG repeats [22,23,24], TPP1 and POT1 bind the single-stranded telomere 3′ overhang as a heterodimer [25,26,27,28,29,30], RAP1 interacts with TRF2 [31], and TIN2 interacts with TRF1, TRF2, and TPP1 to link various subunits of Shelterin together [25,27,32,33]. The CST complex includes CTC1, STN1, and TEN1 [34,35]. They bind the telomere 3′ overhang as a heterotrimer [36], which is structurally similar to the RPA complex [34,37,38,39]. Budding yeast *Saccaromyces cerevisiae* telomere has a sequence of (TG_1–3_)_n_ [40], which is not a perfect repeat, so its telomere protein complex is quite different from those in vertebrates [41]. *Sc*Rap1 binds the duplex telomeric DNA [42,43,44], while the TRF homolog, *Sc*Tbf1, binds subtelomeric TTAGGG repeats [45]. TIN2, TPP1, and POT1 homologs are absent in *S. cerevisiae*, but *Sc*CST (containing Cdc13, Stn1, and Ten1) binds the telomere 3′ overhang and its function is largely conserved as that of mammalian CST [46,47,48]. Telomeres in *T. brucei* have the same TTAGGG repeat sequence as those in vertebrates [3,49,50], and the *T. brucei* telomere complex is more conserved to that in vertebrates than in budding yeast. *T. brucei* has a TRF homolog, *Tb*TRF, that binds the telomeric dsDNA [51], a RAP1 homolog, *Tb*RAP1, that interacts with *Tb*TRF [52], and *Tb*TIF2, which is functionally homologous to TIN2 and also interacts with *Tb*TRF [53]. In addition, a couple of essential DNA polymerases, PolIE and PPL2, have been found to be intrinsic to the telomere chromatin [54,55], and a nonessential protein, TelAP1, has been identified to bind the telomeric DNA [55]. However, *T. brucei* does not seem to have TPP1, POT1, or CST homologs [54]. Recent studies on PolIE indicate that it has similar functions as CST in that it suppresses telomerase-mediated telomere elongation and stimulates the telomere C-strand fill-in process [20,54], although detailed mechanisms are still unclear.

Telomere repeats serve as a docking site for proteins that bind telomeric DNA directly or indirectly. Critically short telomeres recruit insufficient amounts of telomere proteins, exposing the natural chromosome ends and frequently inducing DNA damage responses, including cell growth arrest [56]. Therefore, maintaining a stable telomere length is essential for telomere end protection, and mammalian and yeast cells with critically short telomeres enter replicative senescence [57,58,59,60]. However, conventional DNA polymerases are incapable of replicating linear DNA molecules completely due to their enzymatic properties (they require a template and a primer and only extend DNA at the 3′ ends), resulting in progressive telomere shortening in proliferating cells [4]. In most eukaryotic cells, telomerase, a specialized reverse transcriptase, can synthesize the G-rich telomeric DNA de novo [1,61,62,63], solving this so-called “end replication problem”. Telomerase has both a protein subunit bearing the reverse transcriptase activity and an RNA subunit that provides the template for de novo telomere DNA synthesis [64,65,66]. On the other hand, telomerase-independent telomere maintenance has been observed in several situations. Drosophila telomeres naturally contain retrotransposon arrays, and transposition is the predominant mechanism for telomere maintenance [67,68]. In telomerase-negative ALT cancer cells, DNA recombination serves as the key mechanism of telomere maintenance [69,70]. In addition, telomerase-null yeast cells can continue to proliferate using DNA recombination-mediated amplification of telomeric/subtelomeric repeats as the mechanism of telomere maintenance [71,72]. Therefore, although telomere recombination is normally a risk factor for genome instability, it can serve as an important telomere maintenance mechanism. In addition, in several eukaryotic pathogens that undergo antigenic variation and harbor their major surface antigen genes at the subtelomere (including *Trypanosome brucei* that causes human African trypanosomiasis, *Plasmodium falciparum* that causes malaria, and *Pneumocystis jirovecii* that causes pneumonia in immune-deficient patients), DNA recombination at the telomere and subtelomere can benefit antigenic variation and enhance pathogen virulence [19,73,74,75,76].

In many eukaryotes, telomeres form a heterochromatic structure and suppress expression of genes located in subtelomeric regions [77,78]. Position effect variegation was originally observed in Drosophila, where euchromatic genes are silenced when they are rearranged or translocated close to the heterochromatin [79]. Subsequently, it was found that in *Saccharomyces cerevisiae*, genes located near the telomere are repressed, which is termed the telomere position effect or telomeric silencing [78]. The telomeric silencing phenomenon has been observed not only in yeast [41,80] and Drosophila [81] but also in human cells [82,83,84,85,86,87] and protozoan parasites, including *T. brucei* [88,89,90,91] and *P. falciparum* [92,93]. Telomeric silencing in microbial pathogens can also be an important player regulating pathogen virulence [52,92,93,94,95,96], and telomere proteins have been shown to play critical roles in telomeric silencing [78].

RAP1 is one of the most conserved telomere proteins, whose homologs have been identified in eukaryotes from kinetoplastids to mammals [31,52,97,98,99,100]. RAP1s have been shown to play essential functions in maintaining telomere stability and in telomeric silencing [101,102]. Particularly, functions of *T. brucei* RAP1 in telomeric silencing and telomere stability maintenance are intimately involved in the regulation of antigenic variation [52,103,104,105,106,107], an essential pathogenesis mechanism [108], as the major surface antigen VSG is expressed exclusively from subtelomeric loci [109,110,111,112]. All known RAP1 homologs have an N-terminal BRCA1 C-terminus (BRCT) domain that is frequently identified in proteins involved in the DNA damage response or cell cycle checkpoint [113,114], a central Myb domain that typically binds dsDNA [115,116], and a RAP1 C-terminus (RCT) domain that is a protein–protein interaction domain conserved among all RAP1 homologs [31] (Figure 1). Below, I will discuss RAP1’s functions in telomeric silencing and chromosome end protection, focusing on the similarities and differences between *Tb*RAP1 and yeast and mammalian RAP1 homologs.

## 2. Yeast and Mammalian RAP1 Homologs Are Essential for Telomeric Silencing and Suppress Telomere Recombination

### 2.1. ScRap1 Is a Key Player of Telomeric Silencing in S. cerevisiae

The telomeric silencing phenomenon has been studied extensively in *S. cerevisiae*, and *Sc*Rap1 is the central factor nucleating the silencing effect [41]. *Sc*Rap1 was identified as a DNA binding factor that binds to both transcription activator and repressor elements [97]. It was later shown that *Sc*Rap1 binds the yeast duplex telomeric DNA directly [42,43,44], helps protect the chromosome end [117,118,119], plays a critical role in telomere length regulation [98,120], and is essential for telomeric silencing [121,122].

After the crystal structure of the central region of *Sc*Rap1 was solved, it was clearly shown that *Sc*Rap1 has a Myb-like domain that is nearly identical to its Myb domain (Figure 1), and the Myb/Myb-like region of *Sc*Rap1 is responsible for binding dsDNA with two tandem ACAYYY sequences [43]. The two domains provided considerable flexibility [123,124,125], allowing *Sc*Rap1 to recognize the imperfect telomere repeats with little structural rearrangements or loss of affinity [44]. *Sc*Rap1 is an essential protein, and the C-terminus of its DNA binding domain is indispensable [126]. A recent genomic study comprehensively summarized *Sc*RAP1’s role as a transcription regulator: *Sc*Rap1 activates ribosomal protein and RNR genes and suppresses glycolysis genes and homothallic mating loci [127]. On the other hand, *Sc*Rap1’s effect on subtelomeric gene expression is predominantly repressive [120,121,128,129,130].

*S. cerevisiae* telomeres are the largest *Sc*Rap1 binding sites, which allow an array of *Sc*Rap1 to bind the telomeric dsDNA directly. Telomere-bound *Sc*Rap1, through its interaction with Sir3 and Sir4 silencers [128,129,130,131,132,133,134,135,136,137,138], recruits Sir2 [133,139,140,141,142,143,144,145,146] to the telomere. Subsequently, Sir2 removes acetyl groups from histone tails in a NAD^+^-dependent manner [147,148,149], establishing a heterochromatic structure at the telomere. In addition, Sir3’s broom-adjacent homology (BAH) domain and Sir4 interact with histone H3 and H4 tails [143,150,151,152], which help propagate the heterochromatin from the telomere toward chromosome internal regions [143,153,154,155]. Interestingly, longer telomere repeats in *S. cerevisiae* are associated with stronger telomeric silencing [121], suggesting that more *Sc*Rap1 binding at the telomere can recruit more silencers and establish a more tightly compacted chromatin structure at the telomere. Furthermore, *Sc*Rap1, together with its interacting factors (Rif and Sir proteins), suppresses RNA pol II-mediated telomere transcription and the TERRA level [156]. TERRA was initially identified in several protozoan parasites [157] and subsequently in all eukaryotes examined [158,159,160,161,162]. TERRA has been shown to play important roles in telomere protection, length regulation, and recombination in mammalian cells and yeasts [163,164,165]. TERRA can invade the duplex telomeric DNA and form a three-stranded R-loop structure with an RNA:DNA hybrid [166], and R-loops have a propensity to induce DNA breaks [167,168]. In general, TERRA and telomere R-loops are expected to interfere with the replication machinery and disturb the passage of the replication fork and telomere processing [169]. Therefore, *Sc*Rap1’s role in suppression of TERRA is linked with its role in telomere stability maintenance (see below).

Interestingly, in budding yeast, *Candida glabrata*, that causes opportunistic bloodstream, urinary track, and vaginal infections, telomeric silencing also regulates its virulence [96,170,171]. The NAD^+^-dependent histone deacetylase Sir2 interacts with Sir4, which is recruited to the telomere by *Cg*Rap1 and yKu. Deacetylation of the histone tails by Sir2, in turn, helps establish and propagate the telomeric/subtelomeric silent domain over 20 kb [96]. The *EPA* gene family in *C. glabrata* is located at subtelomeres and encodes epithelia adhesins required for host–pathogen interaction [95]. With normal telomeric silencing, only selected *EPA* genes are expressed. However, under the condition of niacin limitation (where niacin is a precursor of NAD^+^), lowered Sir2 activity leads to weaker telomeric silencing, expression of more *EPA* genes (such as *EPA6*), and enhanced *C. glabrata* adherence to host cells [172].

Telomeric silencing has also been observed in human cells. When a luciferase reporter gene is inserted at a subtelomeric position, it is expressed ~10× lower than when it is inserted at a chromosome internal locus, and longer telomeres induce a stronger silencing effect [83], which depends on histone deacetylation [83]. In addition, expression of a subtelomeric neomycin reporter gene on a Linear Human Artificial Chromosome (L-HAC) is repressed by nearby telomeres and inversely correlated with the telomere length and subtelomeric DNA methylation [173]. Similar to that observed in yeast, human RAP1 can also suppress TERRA expression [174], although the underlying mechanism is unclear. Interestingly, human telomeres can interact with chromosomal internal telomeric sequences via TRF2 [175]. This telomere loop-back can silence genes located over a long distance (a phenomenon termed TPE-OLD), including the *hTERT* gene located ~1.2 Mb away from the telomere [175]. Telomeric silencing appears to also influence the development of FSHD (facioscapulohumeral muscular dystrophy). Sufficient epigenetic alternation of the D4Z4 array located at the chromosome 4 subtelomere (4q35) can lead to abnormal expression of the nearby *DUX4* gene, which is linked to FSHD development [176]. It has been shown that telomeric silencing affects *DUX4* expression [86]. In addition, weaker telomeric silencing can allow more inter- and intra-chromosomal subtelomeric rearrangements of the 4q35 locus [177]. Furthermore, shorter telomeres are correlated with D4Z4 locus hypomethylation, and TPE-OLD regulates *SORBS2* gene expression in FSHD cells [178].

### 2.2. RAP1 Homologs in Higher Eukaryotes Suppress Telomere Recombination

*Sc*Rap1 has essential functions in protecting the chromosome end. Conditional deletion of *Sc*Rap1 leads to yKu70/80, DNA ligase 4, Lif1, and the MRN complex (Mre11/Rad50/Xrs2)-dependent [179,180,181,182,183], non-homologous end-joining (NHEJ)-mediated chromosome end-to-end fusions [117], while the interactions between *Sc*Rap1’s RCT domain with Rif2 and Sir4 are both required for this function [118]. Similarly, human RAP1 can inhibit NHEJ together with TRF2 in vitro [184,185] and help suppress telomere end fusions in senescent cells with short telomeres [186]. In addition, RAP1 homologs have a more conserved function in suppressing homologous recombination (HR) at the telomere. The central region of *Sc*Rap1 inhibits the recruitment of HR proteins independent of yKu, Cdc13, and Rif1/2 [119]. The Rap1 homolog in *Candida albicans* is critical to maintain the telomere length and structure by suppressing telomere recombination [187]. In addition, in methylotrophic yeast *Hansenula polymorpha DL-1*, *Hp*RAP1B (one of the two RAP1 homologs) that binds to the telomere repeats also suppresses telomere recombination [188]. Deletion of mouse RAP1 leads to more homology-directed repair at the telomere (shown as an elevated amount of Telomere-Sister Chromatid Exchanges) [189], as TRF2 and RAP1 suppress PARP1 and SLX4, respectively [190].

## 3. *T. brucei* RAP1 Ensures VSG Monoallelic Expression and Suppresses Telomere Recombination through Unusual Mechanisms

### 3.1. Trypanosoma brucei Undergoes Antigenic Variation to Evade the Host’s Immune Response

*T. brucei* is a protozoan parasite that causes human African trypanosomiasis, which is frequently fatal without treatment. While proliferating in its mammalian host, *T. brucei* stays in extracellular spaces and is immediately exposed to the host’s immune surveillance. However, *T. brucei* sequentially expresses distinct variant surface glycoproteins (VSGs), its major surface antigen, thereby effectively evading the host’s immune response [191]. This antigenic variation is a key pathogenesis mechanism that allows the parasite to establish a long-term infection.

*T. brucei* has a large *VSG* gene pool, including >2500 *VSG* genes and pseudogenes [110], and all are located at subtelomeric regions [111,112]. Most *VSG* genes and pseudogenes are in long *VSG* gene arrays at subtelomeres of mega-base chromosomes that contain all essential genes [112]. Individual *VSG* genes are also found in two-thirds of telomeres of the ~100 mini-chromosomes that predominantly consist of repeat sequences [110,192]. However, at the bloodstream form stage (when *T. brucei* proliferates inside its mammalian host), VSGs are expressed exclusively from bloodstream form VSG expression sites (ESs), which are large polycistronic transcription units with the *VSG* gene at the end within 2 kb from the telomere repeats and the ES promoter 40–60 kb upstream [111]. The Lister 427 *T. brucei* strain used in many research laboratories has ~15 different bloodstream form VSG ESs, all with the same gene organization and ~90% sequence identity [111,112]. However, at any moment, only one ES is fully transcribed by RNA pol I, resulting in a single type of VSG being expressed on the cell surface [193,194]. Monoallelic VSG expression is important for *T. brucei* survival in its mammalian host, as parasites artificially expressing multiple VSGs are more efficiently eliminated by the infected mouse host [195].

VSG switching can occur at the transcription level, where the active ES is silenced while a silent ES is de-repressed (termed in situ switch) [196,197]. In addition, DNA recombination can replace the original active *VSG* sequence with a new one, resulting in expression of a different VSG [196,197,198,199]. HR in *T. brucei* appears to be very active, allowing efficient gene targeting [200,201,202]. HR factors, including RAD51, RAD51-3 (a paralogue of RAD51), and BRCA2, have been shown to be important for normal VSG switching [203,204,205,206]. *T. brucei* lacks the NHEJ machinery [207] but has the microhomology-mediated end-joining (MMEJ) pathway [208,209]. However, whether VSG switching can be mediated solely by MMEJ is unknown. In addition, during a switching event, pieces of various *VSG* donors can be patched together to form a new functional mosaic *VSG* in the active ES [210,211]. Several factors involved in DNA replication, DNA damage repair, and DNA recombination have been shown to suppress VSG switching, including *Tb*ORC1 [212], RECQ2 [213], TOPO3α [214], and RMI1 [215]. On the other hand, inducing DNA double-strand breaks (DSBs) in or immediately upstream of the active *VSG* gene can increase the VSG switching rate ~250 fold, and DSBs in the *VSG* vicinity are a potent trigger of VSG switching [74,75,216,217,218,219]. However, how VSG switching is initiated naturally in *T. brucei* is less clear.

The telomere structure and telomere proteins also influence the VSG switching rate. *T. brucei* cells carrying an extremely short telomere downstream of the active ES have a ~10× higher VSG switching rate compared to cells with longer telomeres (10–15 kb, on average) [220]. The active *VSG* ES-adjacent telomere has been observed to experience frequent truncations during cell proliferation [221]. Presumably, short telomeres have a higher chance to have DNA breaks land in the active *VSG* or nearby, which in turn induces VSG switching. In addition, *T. brucei* telomere proteins (including *Tb*TRF, *Tb*RAP1, *Tb*TIF2, and PolIE) suppress VSG switching by maintaining the telomere integrity and stability, although the underlying mechanisms are not identical [53,54,104,222,223].

### 3.2. Multiple Mechanisms Are Employed to Ensure VSG Monoallelic Expression

Monoallelic gene expression or allelic exclusion has been observed from bacteria to mammals and is important for organism fitness and survival [224,225]. Notable examples include genome imprinting to express one of the two parental alleles, X chromosome inactivation, and random monoallelic expression of autosomal genes in mammals [224,226]. Many monoallelically expressed genes encode cell surface receptors. For example, each αβ T cell expresses one α and one β polypeptide of the T cell receptor, and each human and mouse olfactory sensory neuron expresses only one odorant receptor gene [224]. Feedback signaling involving the gene product (protein) [227] and epigenetic regulation [228] are sometimes employed to achieve monoallelic expression, but detailed mechanisms remain poorly understood.

In *T. brucei*, VSG monoallelic expression is tightly regulated through multiple mechanisms [229,230,231]. First, the subnuclear localization of the active ES is unique. Transcription of the active ES by RNA pol I occurs at a specialized ES body (ESB) located outside of the nucleolus (where RNA pol I transcribes rRNA) [232]. ESB1 has recently been identified to be essential for the active ES transcription and is responsible for recruiting RNA pol I and forming a local, highly SUMOylated focus at ESB [233], where SUMOylation has been shown to positively regulate VSG expression [234,235]. On the other hand, silent ESs are dispersed in the nucleus, away from ESB [236]. Nearly all genes in *T. brucei* are organized in polycistronic transcription units [112,237,238], and the polycistronic transcripts are trans-spliced to have a spliced leader added at the 5′ end of the individual gene transcript [239,240]. Consistently, the active ES associates with the spliced leader gene array in Hi-C analysis that examines chromosome conformation [241], and ESB is located adjacent to one of the two splicing centers shown by immunofluorescence analysis [241]. Presumably, the high-level transcription of the active *VSG* ES is intimately coupled with trans-splicing to improve VSG expression efficiency. Recent studies also showed that VEX1 [242] and VEX2 [243] are essential for VSG monoallelic expression. Both proteins help sustain transcription of the active *VSG* ES in an allelic exclusive manner [242,243]. VEX1 associates with the spliced leader gene array and one of the splicing centers in the nucleus, while VEX2 associates with the active *VSG* ES [241,243]. VEX1 can interact with VEX2, but assembly of the VEX complex relies on RNA pol I transcription [243].

Second, VSG ES transcription is regulated at both initiation and elongation steps. The RNA pol I transcription factor complex, CITFA, has been identified [244]. Two essential CITFA subunits occupy the active ES promoter at a much higher level than at those of silent ESs, and the high promoter occupancy of CITFA is correlated with high levels of RNA pol I occupancy and ES transcription [245], indicating that transcription initiation is different in the active and silent ESs. In addition, transcription elongation along ESs is also regulated, as silent ES promotors are also moderately active, but transcription elongation quickly attenuates after a few kbs, effectively blocking transcription of downstream *VSGs* [246,247].

Third, the chromatin structure plays important roles in VSG ES expression regulation. Although all ESs have ~90% sequence identity [111], silent ESs are packed with nucleosomes, while the active ES is depleted of nucleosomes [248,249,250]. Histone H1 and H3 are required for silencing reporter genes targeted immediately downstream of silent ES promoters, but not the downstream *VSG* genes [251,252]. The *Tb*ISWI complex (including *Tb*ISWI, NLP, FYRP, and RCCP) has been identified to suppress reporter genes at the ES promoters, where *Tb*ISWI has a highly conserved SWI2/SNF2 family ATPase domain and a SANT domain with DNA binding activity [253,254,255]. In addition, histone chaperones FACT (including *Tb*Spt16 [256] and Pob3 [257]), ASF1A, and CAF-1b have all been identified to be required for ES promoter silencing [252]. Besides chromatin remodeling factors involved in ES promoter silencing, TDP1, an architectural HMG chromatin protein, is enriched at the active ES and rDNA and is essential for full-level VSG and rRNA transcription by RNA pol I, presumably by binding DNA directly and excluding nucleosomes [258]. Furthermore, DOT1b, which trimethylates lysine 76 of histone H3, is required for a tight VSG ES silencing [259].

### 3.3. Competition between TbRAP1’s DNA and RNA Binding Activities Is Essential for VSG Monoallelic Expression

As VSGs are expressed exclusively from subtelomeric regions, the telomere structure and telomere proteins have been shown to regulate VSG switching and VSG monoallelic expression. *T. brucei* has the same telomere sequence and terminal 3′ overhang structure as vertebrates [3,49,50,260,261], and the T-loop structure has been observed at both telomeres of the same chromosome [6]. Both the protein and RNA components of *T. brucei* telomerase, *Tb*TERT and *Tb*TR, have been identified, and the telomerase-mediated telomere synthesis is the predominant telomere maintenance mechanism in *T. brucei* [260,262,263,264,265]. As mentioned above, *Tb*TRF [51], *Tb*RAP1 [52], and *Tb*TIF2 [53] are Sheltering-equivalent telomere proteins. Several other *T. brucei* proteins seem to specifically associate with the telomere chromatin but are not homologous to known core telomere proteins in higher eukaryotes, including TelAP1 [55], PolIE [54,266], and PPL2 [54,55,267]. In addition, *Tb*Ku70/80 [268,269] and ORC1 [212] are localized at the telomere.

Although most known *T. brucei* telomere proteins (including *Tb*TRF, *Tb*TIF2, PolIE, and ORC1), when depleted, lead to de-repression of selected *VSGs* up to 10–20-fold [52,53,54,212,266,270], depletion or conditional knockout of *Tb*RAP1 by far results in the most severe *VSG* de-repressing phenotype, where silent *VSGs* are de-repressed up to a thousand-fold, and nearly all *VSG*s in the genome are affected [52,105,106,107].

Examination of the chromatin structure by Formaldehyde-Assisted Isolation of Regulatory Elements (FAIRE) [271] and micrococcal nuclease (MNase) digestion showed that *Tb*RAP1 helps compact the telomeric and subtelomeric ES chromatin structure, although this effect is more prominent in the insect form of *T. brucei* that proliferates in the midgut of its insect vector than in the bloodstream form of *T. brucei* that proliferates in its mammalian host [103]. This observation suggests that *Tb*RAP1-mediated telomeric silencing is an epigenetic effect that relies on modulation of the chromatin structure, which is similar to telomeric silencing in yeast and human, although the underlying mechanism is not fully understood.

Quantitative RT-PCR has been used to measure depression of several genes along the same ES in *Tb*RAP1-depleted cells: a unique *VSG* gene immediately upstream of the telomere repeats, a unique *VSG* pseudogene in the middle of the ES, and a luciferase reporter gene inserted immediately downstream of the ES promoter [52]. As expected, the *Tb*RAP1-mediated silencing is strongest for the *VSG* gene located adjacent to the telomere and weakest for the luciferase gene 40–60 kb upstream [52]. This suggests that the silencing effect is nucleated at the telomere and spreads continuously toward the chromosome internal region, similar to what has been observed in budding yeast at the VII-L telomere with inserted reporter genes [80,272]. Importantly, this observation strongly suggests that *Tb*RAP1’s association with the telomere chromatin is essential for *VSG* silencing.

Different RAP1 homologs use different means to be localized at the telomere. *Sc*Rap1 uses its central Myb and Myb-like domains to bind dsDNA in a sequence-specific manner, as described above [42,43,273]. Similarly, *C. albicans* RAP1 also has both Myb and Myb-like domains and can bind dsDNA directly [187,274]. However, human and *S. pombe* RAP1s do not have any DNA binding activities and are only recruited to the telomere through their interaction with TRF2 and *Sp*Taz1, respectively [31,275]. Sequence analysis showed that *Tb*RAP1 has the conserved BRCT, Myb, Myb-like, and RCT domains (Figure 1), but the sequence homology is pretty weak [52]. Although *Tb*RAP1 interacts with *Tb*TRF [52], *Tb*RAP1 is also located at the telomere in *Tb*TRF-depleted cells [107], indicating that *Tb*TRF is not required for recruiting *Tb*RAP1 to the telomere. This is further confirmed by the observation that the *Tb*RAP1∆Myb mutant still associates with the telomere chromatin, while *Tb*TRF interacts with *Tb*RAP1’s Myb domain [106,107]. This observation strongly suggests that if *Tb*RAP1 Myb has any DNA binding activity, it is not essential for targeting *Tb*RAP1 to the telomere.

Interestingly, *Tb*RAP1 has an _737_RKRRR_741_ (R/K) patch in its Myb-like domain and as part of its bipartite nuclear localization signal (NLS) [106,107]. The *Tb*RAP1 Myb-like domain is required for its interaction with Importin α and its nuclear localization, presumably because Importin α interacts with NLS and helps import *Tb*RAP1 into the nucleus [106]. Surprisingly, the *Tb*RAP1 R/K patch has both dsDNA and ssDNA binding activities, which has been demonstrated by NMR titration and EMSA analyses [107]. These DNA binding activities of *Tb*RAP1 are sequence non-specific and electrostatics based. In addition, *Tb*RAP1 binds longer DNA substrates better than shorter ones, and the shortest dsDNA and ssDNA substrates that *Tb*RAP1 can recognize are ~60 bp and 60 nt, respectively [107]. Importantly, this R/K patch is essential for *Tb*RAP1’s association with the telomere chromatin, as mutating the R/K patch to five alanine residues (5A) removes *Tb*RAP1 from the telomere, which is confirmed by both Chromatin IP and Immunofluorescence analyses [107]. Proteomic studies showed that the two serine residues next to the R/K patch, serine 742 and serine 744, can be phosphorylated in *T. brucei* cells at both the infectious and the insect stages [276,277]. Interestingly, phosphomimetic mutation S742DS744D (2SD) disrupts most of *Tb*RAP1’s dsDNA binding without significantly reducing its ssDNA binding activity, while the S742AS744A (2SA) mutation does not affect *Tb*RAP1’s DNA binding at all [107]. Presumably, phosphorylation of the serine residues brings sufficient negative charges near the R/K patch, which disrupts *Tb*RAP1’s interaction with dsDNA. Importantly, *Tb*RAP1-2SD is no longer localized at the telomere, while *Tb*RAP1-2SA still associates with the telomere chromatin, indicating that the R/K patch-mediated dsDNA binding activity of *Tb*RAP1 is required for its recruitment to the telomere [107]. Furthermore, *Tb*RAP1’s dsDNA binding activity is essential for cell proliferation, *VSG* silencing, and telomere integrity: both *Tb*RAP1 5A and 2SD mutants experience acute growth arrest when the WT complementary allele is conditionally deleted, approximately ten thousand genes are up- or down-regulated in these mutants, and these mutants have an increased amount of DNA damage in the telomeric and subtelomeric regions [107]. Apparently, *Tb*RAP1’s dsDNA binding activity is essential, which is the same as *Sc*Rap1. However, the properties and regulation of *Tb*RAP1 and *Sc*Rap1’s DNA binding activities are drastically different.

Interestingly, *Tb*RAP1 and nuclear phosphatidylinositol 5-phosphatase (PIP5Pase) have been identified in the same protein complex [278,279], where PIP5Pase can dephosphorylate PI(3,4,5)P3. A C-terminally HA-tagged *Tb*RAP1 can bind PI(3,4,5)P3 but not PI(4,5)P2, and depletion of PIP5pase also leads to a dramatic global VSG de-repressing phenotype [278,279,280]. The current model is that *Tb*RAP1’s binding to PI(3,4,5)P3 interferes with its association with the telomere chromatin and disrupts the *Tb*RAP1-mediated *VSG* silencing, while PIP5Pase-mediated dephosphorylation of PI(3,4,5)P3 ensures that *Tb*RAP1’s telomere association is not perturbed. The ability of *Tb*RAP1 to bind phosphoinositol metabolites is unique among RAP1 homologs, and this interaction appears to be important for regulation of *Tb*RAP1’s DNA binding function, although the detailed mechanism is unclear. On the other hand, *Tb*RAP1 does not seem to directly regulate the parasite metabolism, even though RNAseq analysis suggests that *Tb*RAP1 is required for the full-level expression of multiple ribosomal protein genes [105,106,107]. In comparison, *Sc*Rap1 is a transcription regulator for ribosomal protein and glycolysis protein genes [97,127], while mouse RAP1 deficiency leads to decreased activation of PPARα and PGC1α, presumably through regulating the expression of genes encoding these enzymes [281,282].

VSG monoallelic expression has two aspects: silencing all but one *VSG* and expressing the active *VSG* at its full-expression level. It has been estimated that ~10 million VSG molecules are expressed on the *T. brucei* cell surface [193], masking invariant surface molecules from the host’s immune surveillance. To make this dense VSG layer, the active *VSG* gene is transcribed by RNA pol I at a very high level [194], producing *VSG* RNA that represents ~10% of total cell RNA [283,284]. Comparing RNA levels of the same *VSG* gene in cells where it is either active or silent indicates that the active *VSG* RNA level is at least 10,000-fold higher than that of a silent *VSG* [103]. In addition, VSG is an essential protein, as depleting VSG protein by RNAi leads to a pre-cytokinesis arrest [285], and blocking VSG synthesis triggers a general arrest in translation initiation [286]. Therefore, while *Tb*RAP1’s dsDNA binding activity is essential for *VSG* silencing, it is important that *Tb*RAP1 allows the active *VSG* to be expressed at its full level. It turned out that *Tb*RAP1 has another nucleic acid binding activity essential for this function [105].

The NMR structure of the *Tb*RAP1 Myb-like domain has been solved recently [105]. Surprisingly, the N-terminal two-thirds of the Myb-like domain fold into a typical RNA Recognition Motif (RRM) that usually binds RNA or ssDNA [287]. NMR titration and EMSA analyses showed that *Tb*RAP1 RRM binds to RNA but not to ssDNA [105]. Interestingly, *Tb*RAP1 RRM can bind RNA substrates containing the *VSG* 3′UTR but not TERRA [105]. Essentially all *VSG* 3′UTRs have a 9-mer and a 16-mer consensus sequence [110,288], and the 16-mer alone is sufficient for *Tb*RAP1’s recognition [105]. *Tb*RAP1 does not bind a 35 nt RNA oligo with a random sequence, indicating that this RNA binding activity is sequence-specific [105]. On the other hand, *Tb*RAP1 RRM also binds a 170 nt RNA substrate without any *VSG* 3′UTR sequence, suggesting that it can recognize more than one RNA sequence [105]. Aligning *Tb*RAP1 RRM with known RRM domains shows that phenylalanine 655 and phenylalanine 694 are key residues in the conserved RNP2 and RNP1 motifs, respectively, which directly interact with RNA [105]. Indeed, mutating the two phenylalanine residues to alanines, leucines, or glutamines (2FA, 2FL, and 2FQ mutants, respectively) abolishes *Tb*RAP1’s RNA binding activity [105]. Surprisingly, the dsDNA and RNA binding activities of *Tb*RAP1 are mutually exclusive: even though *Tb*RAP1’s dsDNA binding and RNA binding activities are mediated by non-overlapping R/K patch and RRM, respectively, a *Tb*RAP1-dsDNA-RNA ternary complex was not observed in EMSA when the *Tb*RAP1 Myb-like fragment was incubated with both dsDNA and RNA substrates of a similar length [105]. This may be partially due to the fact that the R/K patch also enhances the RNA binding activity of *Tb*RAP1, even though amino acid residues in the R/K patch do not interact directly with the RNA substrate in NMR titration analysis [105].

In vivo, RNA-CLIP analysis shows that *Tb*RAP1 binds the active *VSG* RNA, and this binding also depends on the *Tb*RAP1 RRM domain and the conserved F655 and F694 residues [105]. Unexpectedly, the 5A mutation that disrupts *Tb*RAP1’s association with the telomere chromatin also disrupts the *Tb*RAP1–*VSG* RNA interaction in vivo, even though *Tb*RAP1 RRM alone can bind *VSG* 3′UTR-containing RNA substrates in EMSA and NMR titration analyses [105]. As *Tb*RAP1 only has a moderate affinity to its RNA substrate, this observation suggests that localizing *Tb*RAP1 in the active *VSG* vicinity is important for *Tb*RAP1’s access to and interaction with a high concentration of the active *VSG* RNA. The *Tb*RAP1-5A mutant does not associate with the telomere chromatin [107], so it presumably cannot gain access to the active *VSG* RNA. On the other hand, mutating F655 and F694 residues does not affect *Tb*RAP1’s DNA binding activities, and all three *Tb*RAP1 point mutants (2FA, 2FL, and 2FQ) are localized at the telomere [105], indicating that *Tb*RAP1’s DNA binding and RNA binding activities are not intrinsically dependent on each other.

Most importantly, the *Tb*RAP1-2FA, 2FL, and 2FQ mutants, which only disrupt *Tb*RAP1’s binding to the active *VSG* RNA without affecting its association with the telomere chromatin, lead to a ~50% reduction of the active *VSG* RNA level [105]. In contrast, *Tb*RAP1-5A and ∆DB mutants that lose the DNA binding activities (and do not interact with the telomere chromatin or the *VSG* RNA in vivo) have only a subtle effect on the active *VSG* RNA level (<10%). Therefore, the *Tb*RAP1 RRM domain binds the active *VSG* RNA, and this interaction is essential for the full-level expression of the active *VSG*. However, the *Tb*RAP1–*VSG* RNA interaction does not affect the RNA stability [105], suggesting that this interaction may be transient and has a signaling function. These observations suggest that the competition between *Tb*RAP1’s dsDNA and RNA binding activities are critical for maintaining *VSG* monoallelic expression [105]; on one hand, *Tb*RAP1’s DNA binding activities ensure its association with the telomere chromatin, and hence telomeric silencing, which represses subtelomeric *VSG* expression globally. On the other hand, *Tb*RAP1’s RNA binding activity enables its interaction with the active *VSG* RNA only at the active *VSG* locus where nascent *VSG* RNA is concentrated [232], which competes for *Tb*RAP1’s binding, prevents *Tb*RAP1 from binding dsDNA, and presumably disrupts the *Tb*RAP1-mediated silencing effect locally, hence permitting a full-level expression of the active *VSG*. This model can also explain why *Tb*RAP1 DNA binding-defective mutants only affect the active *VSG* expression subtly [105]: although *Tb*RAP1-5A and ∆DB mutants do not interact with the active *VSG* RNA in vivo, their dissociation from the telomere chromatin already disrupts the *Tb*RAP1-mediated telomeric silencing in the first place, so the *Tb*RAP1–*VSG* RNA interaction is no longer necessary to counteract the silencing effect.

These findings answer the question of how *Tb*RAP1 allows a full expression of the active *VSG* while silencing the other *VSGs* simultaneously. Identification of *Tb*RAP1’s RRM domain and RNA binding activity further brings the investigation of gene expression regulation into a new paradigm, where both dsDNA and RNA binding activities of the same protein and the competition between the two activities are essential for monoallelic gene expression.

It is worth noting that VSG monoallelic expression employs critical allelic exclusion mechanisms. As described above, VEX2 is associated with ESB and repels silent ES from gaining access to RNA Pol I [241,243]. In addition, the total amount of *VSG* mRNA appears to be fixed, as expressing an ectopic *VSG* gene leads to downregulation of the endogenous *VSG* expression [286,288,289,290,291]. This is possibly regulated through the 16-mer *VSG* 3′ UTR consensus sequence [288], which can be bound by CFB2 [292,293] and modified with m6A [294], both important for *VSG* mRNA stability. However, adding stop codons in the *VSG* gene ORF leads to dramatically increased total *VSG* mRNA levels, suggesting that translation of *VSG* mRNA is actually monitored to regulate VSG expression [295]. Regardless of the underlying mechanism, the need to keep a fixed amount of total expressed VSG can explain why *Tb*RAP1 RNA binding-defective mutants also cause de-repression of silent *VSGs* at slower kinetics than *Tb*RAP1 DNA binding-defective mutants [105], as a dramatic decrease in the active *VSG* expression can signal for increased expression of silent *VSGs*. Alternatively, it is also possible that the *Tb*RAP1–*VSG* RNA interaction may directly signal for allelic exclusion of VSG expression, and loss of this interaction allows de-repression of silent *VSGs*.

### 3.4. TbRAP1 Helps Maintain Telomere Stability and Suppresses VSG Switching

As a telomere protein, *Tb*RAP1 is also essential for maintaining telomere integrity and stability: depletion of *Tb*RAP1 by RNAi results in more γH2A (deposited to chromatin at the DNA damage site [296]) and RAD51 associated with the telomere and subtelomere chromatin when analyzed by Chromatin IP, an increased amount of DNA breaks at the subtelomeres when analyzed by Ligation-Mediated PCR, and an elevated VSG switching rate with more *VSG* gene conversion events when analyzing the status of markers inserted in the active ES [104]. In addition, *Tb*RAP1 point mutants, including 5A and 2SD that disrupt its DNA binding/telomere localization, and 2FQ, 2FL, and 2FA that disrupt its RNA binding, all lead to an increased amount of telomere- and subtelomere-associated γH2A [105,107]. Interestingly, *Tb*RAP1 also suppresses the TERRA level [104], which is an important mechanism of how *Tb*RAP1 maintains telomere integrity and stability.

TERRA was originally detected in *T. brucei*, which contains the G-rich telomere repeat sequence, and only a small portion of it is poly-adenylated [157]. In *T. brucei*, TERRA appears to only be transcribed from the active *VSG* ES-adjacent telomere by RNA pol I, presumably as a read-through product [104,270]. In the G1 phase, most *T. brucei* cells have a single nuclear TERRA focus. As cells progress through the cell cycle, most cells have 2 or 3 nuclear TERRA foci in S or G2/M phases [270]. Depletion of *Tb*TRF leads to more cells with fewer number of TERRA foci throughout the cell cycle [270]. Both *Tb*RAP1 and *Tb*TRF suppress the TERRA level [104,270]. In addition, depletion of *Tb*RAP1 or *Tb*TRF also results in more telomeric R-loops (TRL), an increased amount of telomere DNA damage, and an elevated VSG switching rate [104,223,270]. Overexpression of an ectopic allele of RNase H1 that degrades RNA in the RNA:DNA hybrid [297,298] can suppress the increased TRL level, more telomeric DNA breaks, and higher VSG switching rate phenotypes in *Tb*RAP1/*Tb*TRF-depleted cells [104,270], confirming that suppressing TERRA and TRL levels is a major underlying mechanism of how *Tb*RAP1 and *Tb*TRF help maintain telomere/subtelomere stability. Therefore, TERRA expression regulation is directly linked to telomere stability through *Tb*RAP1 and *Tb*TRF. However, this regulation, mediated by *Tb*RAP1 and *Tb*TRF, is presumably not at the transcription initiation level, as TERRA is transcribed by RNA pol I downstream of the active *VSG* ES [104,270], with its promoter 40–60 kb upstream [111]. *Tb*TRF binds TERRA directly using its C-terminal Myb domain [270] that also binds the duplex TTAGGG repeats [51], which may contribute to its role in TERRA suppression. *Tb*RAP1 does not bind TERRA [105], but TERRA species are much longer in *Tb*RAP1-depleted cells [104], suggesting that *Tb*RAP1’s binding to the telomere or *Tb*RAP1-mediated telomere chromatin compaction deters RNA pol I transcription elongation.

## 4. Discussion

In summary, it is clear that the fundamental telomere functions in different eukaryotes are conserved, as telomere proteins help protect the chromosome ends and maintain telomere integrity and stability, and genes located in subtelomeric regions are frequently silenced by the nearby telomeric chromatin. However, telomere proteins, including their identities, sequences, functional domains, and mechanisms of their action, frequently have species-specific features.

Specifically for RAP1 homologs, their roles in maintaining telomere integrity and stability are conserved (Figure 2), and RAP1 homologs have been shown to suppress HR in the telomere vicinity: *Tb*RAP1 suppresses subtelomeric *VSG* gene conversion events, and the recombinase RAD51 is recruited to telomeric/subtelomeric DNA damage sites when *Tb*RAP1 is depleted [104]. *Sc*Rap1 suppresses recruitment of the Mec1-Ddc2 checkpoint complex and the Rad52, Mre11, and RPA recombination machinery to the telomere [119,299]. Mammalian RAP1 also coordinates with TRF2 to suppress HR-mediated telomere deletion and fusions [189,190], even though they are not essential [281,300,301]. *Sc*Rap1 also suppresses NHEJ at the telomere [117,118]. Mammalian RAP1 has been reported to suppress NHEJ in vitro [184,185] and suppress telomere fusions in cells with short telomeres [186]. *T. brucei* does not have the NHEJ machinery [207], and no telomere end-to-end fusions have been reported in *Tb*RAP1-depleted cells. In addition, *T. brucei,* yeast, and mammalian RAP1s all suppress TERRA [104,156,160,174], which in turn helps maintain telomere stability. This also links RAP1’s telomeric silencing effect with its chromosome end protection function. However, *Sc*Rap1 presumably suppresses TERRA at the transcription initiation step, while *Tb*RAP1 apparently suppresses RNA pol I transcription elongation, even though both may involve chromatin structure remodeling.

RAP1 homologs’ functions in gene transcription regulation appear more divergent (Figure 3). *Sc*Rap1’s transcription activation function depends on its binding to the ribosomal protein gene promoter [97], and *Sc*Rap1 also represses cryptic or divergent transcription to ensure the transcription goes in the correct direction [302,303]. On the other hand, *Sc*Rap1-mediated telomeric silencing depends on its recruitment of Sir proteins to the telomere and a heterochromatic telomere structure [41]. Mouse RAP1 also binds multiple chromosomal internal sites and regulates nearby gene expression [281]. In addition, mouse RAP1 exhibits a repressive effect on the expression of genes located at subtelomeres [281]. Furthermore, cytoplasmic RAP1 interacts with IκB [304], the inhibitory protein partner of NFκB [305]. The RAP1–IκB interaction releases NFκB, which permits NFκB to be activated as a transcription activator [304]. Hence, mammalian RAP1 has an additional extra-telomere gene expression regulation function.

*Tb*RAP1 is clearly required for silencing subtelomeric *VSG* genes, which appears to involve chromatin remodeling (Figure 3, top) [103]. However, *Sc*Rap1-mediated silencing affects RNA pol II access to the subtelomeric gene promoter, while *Tb*RAP1 likely blocks RNA pol I transcription elongation, presumably as a structure protein intrinsic to the telomere chromatin. In addition, RNAseq analysis showed that conditional knockout of *Tb*RAP1 also caused a ~35% decrease in the RNA level of a number of ribosomal protein genes [106,107], although it has not been reported whether these genes are within the same polycistronic transcription units and whether *Tb*RAP1 directly binds their promotor. Interestingly, *Tb*RAP1 also helps ensure full-level expression of the active *VSG,* which is not due to any canonical transcription activation function (Figure 3, bottom) [105]. Rather, through competing with *Tb*RAP1’s dsDNA binding activity, its RNA binding activity inhibits its silencing effect [105]. Both the RRM domain and RNA binding activity are unique in kinetoplastid RAP1s but absent in yeast or mammalian RAP1 homologs [105]. A recent finding indicates that many human transcription factors have not only DNA binding but also arginine-rich motif (ARM)-mediated RNA binding activities, which help fine-tune transcription regulation [306]. Therefore, many more proteins than we previously knew seem to have both DNA and RNA binding activities that play important roles in gene expression regulation. It would be intriguing to see whether yeast and mammalian RAP1 homologs have similar ARM domains and RNA binding activities.

Identification and characterization of functions of *T. brucei* telomere proteins revealed that the essential functions of telomere proteins are largely conserved, even though homologs from kinetoplastids to mammals only have limited sequence homology within functional domains, which shed light on telomere protein evolution. Importantly, unique features of *Tb*RAP1 DNA and RNA binding activities and their regulations have been identified that are absent from mammalian RAP1s. Therefore, *Tb*RAP1, an essential telomere protein, is also a promising target for future development of anti-parasite agents.

## Figures and Tables

**Figure 1 biomolecules-14-00067-f001:**
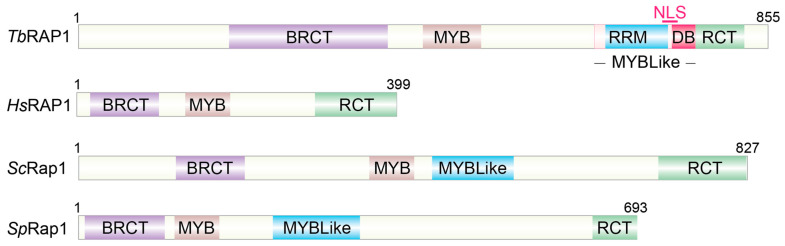
Domain structures for four RAP1 homologs: *Tb*RAP1, *Trypanosoma brucei* RAP1; *Hs*RAP1, *Homo sapiens* RAP1; *Sc*Rap1, *Saccharomyces cerevisiae* Rap1; *Sp*Rap1, *Schizosaccharomyces pombe* Rap1. BRCT, BRCA1 C-terminus; RRM, RNA Recognition Motif; DB, DNA binding; MYBLike, Myb-like; RCT, RAP1 C-terminus. Numbers stand for amino acid positions.

**Figure 2 biomolecules-14-00067-f002:**
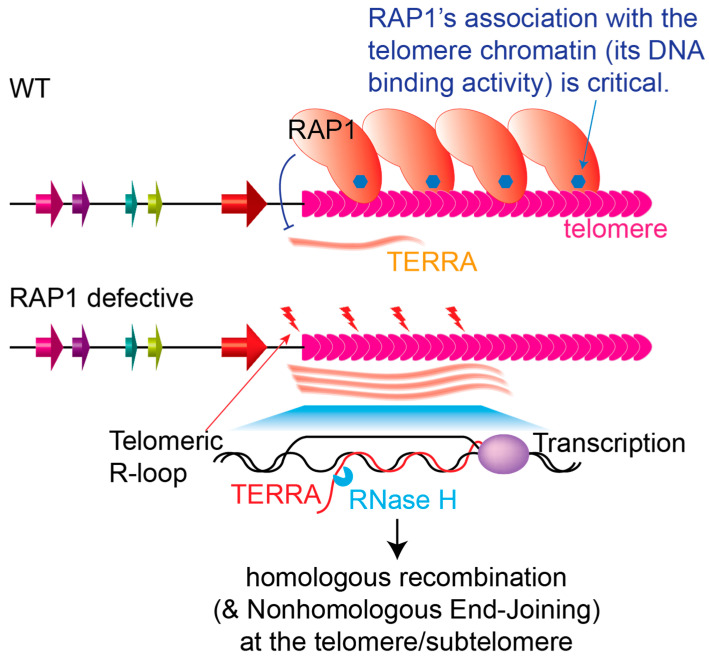
RAP1 homologs suppress telomeric homologous recombination. RAP1 homologs have been shown to suppress the TERRA level, where TERRA can form telomeric R-loops that induce telomere instability, and homologous recombination is a major DNA DSB repair pathway, particularly in *T. brucei*. Red bolts represent DNA damage. Blue hexagons in RAP1s represent DNA binding or the critical domain required for its recruitment to the telomere. Budding yeast and human RAP1s also suppress NHEJ under special conditions (not shown).

**Figure 3 biomolecules-14-00067-f003:**
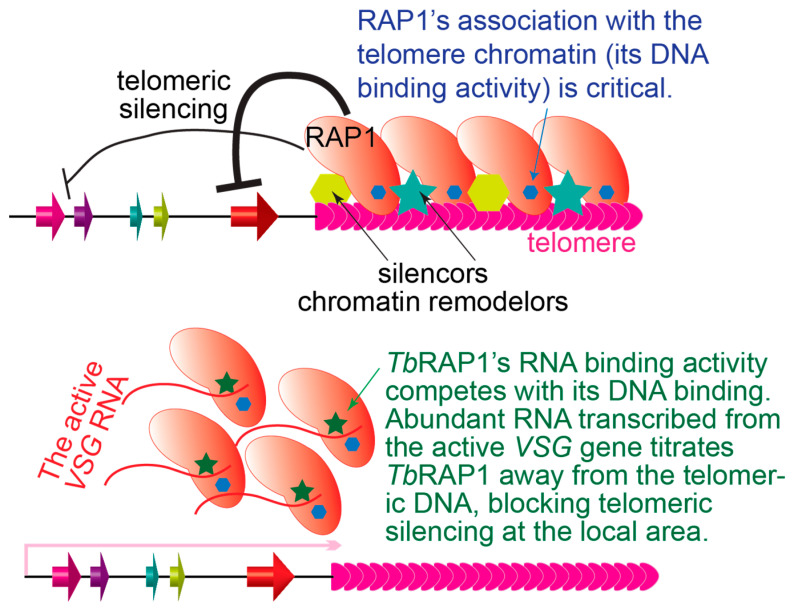
RAP1 homologs are essential for telomeric silencing. Top: RAP1s help establish/maintain telomeric silencing, presumably by modulating the chromatin structure through recruiting silencers and chromatin remodelers. Curved lines with a short bar at the end represent the silencing effect. The thickness of the lines represents the strength of the silencing effect. Blue hexagons in RAP1 homologs represent DNA binding or the critical domain required for its recruitment to the telomere. Bottom: *Tb*RAP1 has a unique RNA binding RRM domain (represented by a green star), and the competition between RNA and DNA binding ensures that the active *VSG* is not suppressed by *Tb*RAP1.

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
