# Peer review of "Unwrap RAP1’s Mystery at Kinetoplastid Telomeres"

_biomolecules, 2024, doi:10.3390/biom14010067_

Round 1

Reviewer 1 Report

Comments and Suggestions for Authors

The manuscript presents a thorough review, discussing the crucial role of telomeres, located at the chromosome end, in maintaining genome integrity and chromosome stability across various organisms. The review emphasizes the conservation of telomere proteins in protecting chromosome ends, suppressing DNA damage response machineries, and preventing nucleolytic degradation. Additionally, it highlights the repressive epigenetic effect of specialized telomere structures on gene expression in subtelomeres, leading to telomeric silencing. This silencing phenomenon extends to affecting the virulence of microbial pathogens undergoing antigenic variation/phenotypic switching. The review delves into the specific functions of RAP1 homologs, particularly in telomeric silencing and the suppression of Telomere Repeat-containing RNA (TERRA), linking these roles to telomere stability maintenance. While the functions of RAP1s in suppressing telomere recombination are conserved across organisms, the review acknowledges that the underlying mechanisms of RAP1-mediated telomeric silencing exhibit species-specific features. The focus of the review is on the functions of Trypanosoma brucei RAP1 in suppressing telomeric/subtelomeric DNA recombination and regulating monoallelic expression of subtelomerelocated major surface antigen genes. The review promises a comparative analysis of common and unique mechanisms among RAP1 homologs and discusses their implications. The manuscript exhibits exceptional writing quality. The author, recognized as a prominent authority in the field, provides a comprehensive review that effectively encapsulates the latest advancements in telomere biology. The content notably enhances readers' comprehension of telomere biology in general and the distinctive properties of RAP1 in Trypanosoma brucei. 

Author Response

I thank the reviewer for a careful review. I am grateful for the reviewer's positive comments.

Reviewer 2 Report

Comments and Suggestions for Authors

The review "Unwrap RAP1’s mystery at kinetoplastid telomeres" by Bibo Li describes the multiple functions of RAP1 telomeric protein from distinct organisms from kinetoplastid through yeasts to the human. The major attention is directed to the kinetoplastid RAP1 protein and manuscript contains very interesting information about telomeric silencing of genes expression that is realized in the regulation of expression of variant surface glycoproteins of T.brucei. 

The review will be interesting for the readers and provides a comprehensive analysis of distinct properties of RAP1 proteins. 

However, I think that it will be more comfortable for readers to percept the information with a help of illustrative material. The review contains only 1 figure with the schematic representation of domain organization of RAP1 proteins from different organisms. I ask the author to add more figures which will illustrate the mechanisms of RAP1 action in cells.

Author Response

I thank the reviewer for a careful and constructive review. I have added two figures summarizing RAP1 homologs' functions in telomere stability maintenance (Fig. 2) and in telomeric silencing (Fig. 3). I hope this addition is satisfactory to the reviewer.